# Classification of Cardiomyopathies from MR Cine Images Using Convolutional Neural Network with Transfer Learning

**DOI:** 10.3390/diagnostics11091554

**Published:** 2021-08-27

**Authors:** Philippe Germain, Armine Vardazaryan, Nicolas Padoy, Aissam Labani, Catherine Roy, Thomas Hellmut Schindler, Soraya El Ghannudi

**Affiliations:** 1Department of Radiology, Nouvel Hopital Civil, University Hospital, 67091 Strasbourg, France; aissam.labani@chru-strasbourg.fr (A.L.); catherine.roy@chru-strasbourg.fr (C.R.); soraya.elghannudi-abdo@chru-strasbourg.fr (S.E.G.); 2ICube, University of Strasbourg, CNRS, 67091 Strasbourg, France; vardazaryan@unistra.fr (A.V.); npadoy@unistra.fr (N.P.); 3IHU, 67091 Strasbourg, France; 4Division of Nuclear Medicine, Mallinckrodt Institute of Radiology, Washington University School of Medicine, Saint Louis, MO 63110, USA; thschindler@wustl.edu; 5Department of Nuclear Medicine, Nouvel Hopital Civil, University Hospital, 67091 Strasbourg, France

**Keywords:** cardiomyopathy, deep learning, transfer learning, convolutional neural network, Grad-CAM

## Abstract

The automatic classification of various types of cardiomyopathies is desirable but has never been performed using a convolutional neural network (CNN). The purpose of this study was to evaluate currently available CNN models to classify cine magnetic resonance (cine-MR) images of cardiomyopathies. Method: Diastolic and systolic frames of 1200 cine-MR sequences of three categories of subjects (395 normal, 411 hypertrophic cardiomyopathy, and 394 dilated cardiomyopathy) were selected, preprocessed, and labeled. Pretrained, fine-tuned deep learning models (VGG) were used for image classification (sixfold cross-validation and double split testing with hold-out data). The heat activation map algorithm (Grad-CAM) was applied to reveal salient pixel areas leading to the classification. Results: The diastolic–systolic dual-input concatenated VGG model cross-validation accuracy was 0.982 ± 0.009. Summed confusion matrices showed that, for the 1200 inputs, the VGG model led to 22 errors. The classification of a 227-input validation group, carried out by an experienced radiologist and cardiologist, led to a similar number of discrepancies. The image preparation process led to 5% accuracy improvement as compared to nonprepared images. Grad-CAM heat activation maps showed that most misclassifications occurred when extracardiac location caught the attention of the network. Conclusions: CNN networks are very well suited and are 98% accurate for the classification of cardiomyopathies, regardless of the imaging plane, when both diastolic and systolic frames are incorporated. Misclassification is in the same range as inter-observer discrepancies in experienced human readers.

## 1. Introduction

Machine learning, computer vision, and deep learning are areas that have grown explosively over the last 10 years. For image processing, most of the work concerns supervised learning with a convolutional neural network (CNN) [1]. Performances of CNN networks for the classification of medical images were demonstrated to be at least on par with if not superior to that of specialist practitioners in several fields [2], and commercial AI-powered medical imaging applications are already available.

An exhaustive review of image-based cardiac diagnosis with machine learning was recently published [3]. Most deep learning (DL) studies performed on computed tomography (CT) data were devoted to calcium scoring, coronary artery disease prognosis, and functional coronary stenosis detection using plaque or fractional flow reserve quantification [4]. Cine-MR studies focused on left-ventricular (LV) segmentation allow the automatic quantification of cardiac volumes and function and are aimed at replacing traditional tedious manual contouring by fully conventional networks with encoder–decoder structure (e.g., U-Net) [5,6].

In addition to studies focused on the myocardial region of interest, using texture analysis, DL cardiac disease classification from global cardiac images has been reported for left-ventricular (LV) hypertrophy by cardiac ultrasound [7,8], perfusion defect by single-photon emission tomography (SPECT) [9], cardiac involvement in sarcoidosis by ^18^F-fluorodeoxyglucose positron emission tomography (PET) [10], and diagnosis of amyloidosis from late-gadolinium-enhanced images [11]. Using cine-MR images, DL was found efficient to improve mutation prediction in hypertrophic cardiomyopathy (HCM) [12].

Diagnosis of hypertrophic and dilated cardiomyopathies (DCM) is important since they have a poor prognosis and require careful treatment and monitoring. Their diagnosis is based on the quantification of LV volumes, myocardial wall thickness, and LV ejection fraction [13]. Cine-MR is well suited and largely used to allow these measurements. A challenge was organized in 2017 to classify cine-MR images of 150 patients, including healthy, HCM, and DCM, but approaches relied on conventional index extraction after cardiac structure segmentation or on myocardial radiomics extraction, followed by random forest or support vector analysis, and not on full image CNN analysis [3]. The aim of our study was to determine performances of commonly available CNNs to differentiate normal, HCM, and DCM using standard cine-MR sequences, without explicit, analytic, cardiac chamber and wall thickness measurements and with a view to the probable gradual deployment of these techniques in future imaging systems.

## 2. Materials and Methods

### 2.1. Study Population

This retrospective study was registered and approved by the Institutional Review Board of our university hospital. All datasets were obtained and deidentified, with waived consent, in compliance with the Institutional Review Board. Cine-MR exams of 534 patients, performed between 2010 and 2019, were retrospectively studied. Study selection was made on the basis of visual diagnosis, by a practitioner with 25 years of experience, reviewing and retaining cine-MR images with the characteristic appearance of normal, hypertrophic, or hypokinetic dilated cardiomyopathy. Table 1 summarizes the study population.

In HCM, the measurement of the diastolic wall thickness was carried out occasionally, in case of doubt, to ensure that myocardial hypertrophy indeed exceeded 15 mm [14]. Most HCM cases fit with the classical asymmetric septal hypertrophy pattern with or without systolic obstruction.

In the DCM group, only presumed primitive DCM forms were selected, i.e., cases with segmental wall thinning or post-gadolinium late enhancement ischemic pattern were withdrawn. Patients with noncompaction were excluded, but septal dyskinesia without septal late enhancement, suggestive of left bundle branch block, was included. Criteria used in case of doubt were diastolic LV enlargement exceeding 63 mm immediately basal to the tips of the papillary muscles (reference normal cutoff was 60 mm in the Framingham Heart Study Offspring cohort [15], while 62–63 mm was reported in the meta-analysis of Kawel Boehm et al. (Table 7, [16])) and a visually estimated LV ejection fraction <40%.

One to six cine sets (mean 2.25 ± 1.28) were selected for each patient, taking into account only typical pathological features for cardiomyopathies. For example, if hypertrophy was limited to the basal septum and not to the anterior wall, only four-chamber and basal ventricular short-axis views were included and not the vertical long-axis view.

### 2.2. Cine-MR Acquisitions

All images were obtained at 1.5 T (three Siemens and one Philips scanner). Only steady-state free precession (SSFP) cine sequences were analyzed with TE/TR in the range 1.6/3.5 ms, slice thickness in the range 6 to 8 mm, and 8–32 cardiac coil elements. Trigger time corresponding to end-systole was visually selected (smallest LV dimension). Imaging planes were vertical long axis with left-sided or right-sided apex (VLAl, VLAr), four-chamber view (4C), and short-axis view (SA). A summary of the cine set statistics is listed in Table 1.

### 2.3. Image Preparation

Digital imaging in communications in medicine (DICOM) files from the cine studies selected on the picture archiving and communications system (PACS) of our hospital were exported to a custom-made preparation software (Visual C), illustrated in Figure 1, in order to perform (1) deidentification of all data related to the patient and to the institution, (2) bilinear resampling, to obtain a normalized homogeneous pixel size of 1.5 mm, (3) gray level windowing, focused on the cardiac region of interest, and (4) selection of the diastolic and systolic frames in the cine set. Finally, three pairs of TIFF images (cropped to 128 and 160 pixels large + full view at 256 pixels large) and one pair of raw bitmaps (without any rescaling) were stored. The attribution of the categories ‘orientation plane’ and ‘pathology’ was carried out at the same time and saved in the label file.

### 2.4. Deep Learning Process

CNN implementation was performed in Python 3.7.6, using the Keras library and TensorFlow backend. According to the classical DL method [17], several reference base models, pretrained on the Imagenet database, were used: VGG16 [18], ResNet50V2, InceptionResNetV2, and DenseNet201. Fine-tuning was applied on the last layers. Base models were followed by a fully connected layer module: Flatten, Dense 256, Dropout 0.25, Dense 128, Dropout 0.35, Dense 64, Dropout 0.40, and finally output Softmax activation layer. Data were randomly shuffled and split into training, validation, and test groups (in the case of a double split process); therefore, the same input was never in two distinct groups. Training was done with a batch size of 32, number of epochs of 100, optimizer SGD, LR of 10^−4^, and categorical cross-entropy as loss function. Data augmentation was applied during training with up to 0.15 zoom range, 20° rotation, and 15% height and width shift range. Model trimming was limited to a few alterations of the number of epochs and of dropout values in the head model. Thus, since adjustments of hyperparameters were minimal, information leaks should in principle be almost absent; hence, nested cross-validation was not performed. Two versions of the model were tested: (1) VGG-single with one input corresponding to the diastolic or systolic frame, and (2) VGG-concat with two inputs supplied by both diastolic and systolic frames, as illustrated in Figure 2.

With chosen parameters, no overfitting was observed. Specific models for orientation plane and for pathology were trained through a sixfold cross validation process, randomly taking one-sixth of the whole inputs as a validation set for each training (i.e., 1000 trained and 200 validated each time). Lastly, in order to test the validity and the generalization of the algorithm, a double split process was performed with a separate hold-out set of data (60% training, 20% validation, and 20% hold-out test data). Performance metrics (validation loss and validation accuracy) resulted from the average of the six training sets performed over 100 epochs. In this way, all misclassified inputs could be stored and visually inspected in an attempt to understand the source of errors. Confusion matrices were used to determine the nature and the rate of misclassification.

### 2.5. Independent Reader Analysis

Diastolic and systolic images of the VGG-concat model, comprising 904 training inputs and 227 validation inputs, were analyzed blindly by a cardiologist and by a radiologist unaware of the image set. Moreover, these images were also reread blindly by the cardiologist who had carried out the initial labeling, more than 4 months before.

Lastly, a complementary series of 795 inputs, previously unseen by the model, were tested separately.

### 2.6. Saliency Maps

Class activation maps were visualized thanks to the Grad-CAM algorithm [19]. From the final convolutional layer in the network, Grad-CAM examines the gradient information flowing into that layer, in order to identify the most contributive pixels involved for each class. The output of Grad-CAM is a heatmap visualization for a given class label.

### 2.7. Evaluation and Statistical Analysis

Two metrics were evaluated for the various models: cross-validation loss and accuracy. Confusion matrix was used to identify the nature of discrepancies between the assigned label and the predicted class. Chi-square test was used for testing relationships between categorical variables, and comparisons between quantitative data or scores were performed with ANOVA or with Student’s *t*-test. Statistical analyses were performed using MedCalc 12.1.4 (MedCalc Software, Ostend, Belgium).

## 3. Results

### 3.1. Classification According to the Four Orientation Planes

Results of the average sixfold cross-validation for the classification according to the orientation plane, with 160 × 160 pixel frames, are listed in Table 2. Test accuracy was >0.998 with only one error in the dual-input model and fewer than 4/1200 input misclassifications for the single-input models. No significant difference was seen between models.

### 3.2. Classification According to the Pathology

The different pretrained base models tested provided quite similar results (less than 4% difference in test accuracy), but InceptionResNetV2 and VGG16 turned out to be the best and were on par. Results listed here were obtained with VGG16 and are summarized in Table 2. For the VGG-single model, the average sixfold cross-validation accuracy was 0.961 ± 0.011 for diastole and 0.952 ± 0.012 for systole (ns). The VGG-concat model based on diastolic and systolic frame pairs outperformed the single-frame model. Cross-validation loss was twofold lower (0.078 ± 0.038, *p* < 0.0036), and cross-validation accuracy was 2–3% (absolute value) better (0.982 ± 0.009, *p* < 0.016), as compared with the single-frame models. The double-split model with separate 20% hold-out test group provided 0.974 ± 0.011 test accuracy. Since the study population was not homogeneous in each class (e.g., regarding gender and age), stratified analysis was performed. According to sex, accuracy for the hold-out test group was 0.932 for male (57% of cases) and 0.883 for female (43%). According to age, we found 0.889 in patients <46 years old (32% of cases), 0.879 in patients 45–62 years old (33%), and 0.907 in patients >62 years old (35%). These differences were interpreted as related to the number of cases studied in each subgroup.

Lastly, the additional analysis carried out on 795 supplementary inputs, never seen before by the model, showed 33/795 errors, i.e., an accuracy of 0.958.

### 3.3. Analysis of Misclassified Cases

Summed results from the six confusion matrices obtained through the sixfold cross-validation training (scanning the whole samples) are listed in Table 3.

With the single-frame model, 47/1200 inputs (3.9%) were erroneously recognized in diastole and 54/1200 (4.5%) were misclassified in systole, with most errors resulting from the wrong classification of normal cases as hypertrophic or dilated cardiomyopathy. The dual-frame concatenated model outperformed both VGG single models with only 22/1200 errors (1.83%, *p* < 0.0008), homogeneously distributed among pathology and view. The double-split experiment with a 20% set of hold-out data provided similar results: 12/240 (5.0%) misclassifications for diastole and systole and 4/240 (1.7%) errors for concatenated inputs. Visual inspection of misclassified inputs showed that errors frequently corresponded to cases which could retrospectively be deemed questionable (for example, localized apical or septal hypertrophy).

### 3.4. Comparison with Human Reader Classification

The classification of a 227-input validation group, carried out by an experienced radiologist and cardiologist, led to a similar number of discrepancies: seven and eight for practitioners vs. eight for the dual-input model, showing that the algorithm performance was on par with that of human readers. The misclassification made by the two human observers concerned the same image in 1/15 cases only. Similarly, the errors made by the algorithm concerned the same image as for human observers in only one case (for both readers). In the event of a discrepancy between the algorithm and the human, the latter was right in a little more than half of the cases. Blind rereading by the cardiologist who carried out the initial labeling showed 3/227 discrepancies corresponding to borderline images, with poorly defined characteristics.

### 3.5. Influence of Image Preparation Parameters on Classification Accuracy

The influence of the image matrix size on the results is reported in Table 4. Significantly higher performances were observed when the image matrix was centered on the cardiac region of interest (128 or 160 matrix size). The raw images with no normalization of the pixel size or level windowing adapted to the cardiac region produced weaker results, which proves the usefulness of the preliminary step of preparing the images before CNN training.

### 3.6. Analysis of the Saliency Maps

Saliency maps reveal the pixel areas responsible for classification. When the correct class is identified, the active zone covers almost entirely the left-ventricular region in the image corresponding to the ground-truth class (Figure 3).

A systematic review of all cases corresponding to the correct ground-truth class revealed 19/1178 observations with an inappropriate active pixel area (located outside of the heart). This means that classification was correctly performed due to unintended features.

In situations where misclassification occurred, heatmap inspection revealed that the main source of error (half of the cases: 11/22) was related to the fact that the extracardiac location caught the attention of the network, as illustrated in Figure 4.

Errors between distinct heart structures explained 6/22 observations, an erroneous predicted class resulting from an extracardiac structure occurred in 3/22, and an erroneous predicted class resulting from no cardiac structure seen at all occurred in 2/22.

## 4. Discussion

The current study provides a unique framework of the concept that applying CNN in cine-MR may contribute to optimizing the identification and characterization of different cardiomyopathy disease entities, because CNN features likely carry important prognostic and therapeutic information. In the present study, using a dual-input CNN model, we obtained 98% accuracy in classifying normal heart, HCM, and DCM from 1200 diastolic and systolic cine-MR frames. Only 22 misclassifications were observed, homogeneously distributed across frame orientation and pathological classes. The external validation study demonstrated a similar number of misclassifications between the algorithm and experienced radiologist and cardiologist. The low rate of errors for the diagnosis of pathology (1.8%) suggests the possibility to consider implementation of such algorithms, providing classification with heat maps, within medical imaging devices.

Our results outperform those using echocardiography. Madani et al. reported a test accuracy of 81% in the diagnosis of LV hypertrophy [7], and Zhang et al. reported an area under the receiver operating characteristic curve of 0.93 for HCM [8]. For view identification, which is an important preliminary step before automatic cardiac chamber segmentation and quantification, only one classification error was found (accuracy 0.998). By comparison, view identification accuracy was 96% [8] and 94% [7] in echocardiographic studies (with weaker image quality than cine-MR).

### 4.1. CNN Models

Classical deep learning methods [17] were used in the present study, by adapting the popular, quite simple, and freely available VGG model. This CNN network is recognized for its good performance (winner of the ImageNet Large-Scale Visual Recognition Challenge in 2012–2013), but the training time from scratch is computationally expensive, which is why we used the transfer learning technique, followed by fine-tuning on the last layers of the model. Thus, feature maps could be adapted for our cine-MR images. This model seems to be well generalizable, for example, with images obtained after gadolinium injection (mostly short axis), comprising pleural or pericardial effusion, or coming from different kinds of MR scanners. Other base models tested did not provide better results, and this is in line with results reported for cardiac short-axis slice range classification [20]. In a large, multicenter study, Betancur et al. [9] used standard Convnet with three feature extraction units for prediction of obstructive coronary artery disease by SPECT. The Inception-V3 network was used to identify cardiac involvement in sarcoidosis by FDG-PET [10].

### 4.2. Importance of Data Preparation

The frame preparation step, including manual cropping of the cardiac region, rescaling to standardize the resolution to 1.5 mm/pixel, and level windowing adapted to the cardiac region, was useful in our study, since classification accuracy was 5% lower (absolute value) when using DICOM raw bitmaps instead. However, this preliminary step requires non-negligible additional working time.

### 4.3. Sources of Human Errors

The ground-truth class cannot be perfectly defined, and this is a common source of potential error, inherent to all studies of this type. Management of a quite large number of observations, requiring sustained attention, led to some labeling errors, which are difficult to avoid completely. Approximately 20 such errors (orientation, pathology, and systolic phase) were identified and corrected during the numerous steps of this work. Moreover, classification discrepancies appeared upon blind rereading of a validation group by an expert radiologist and cardiologist (7/227 and 8/227) and even by the cardiologist who carried out the initial labeling (3/227). Discussion between colleagues showed that those cases were questionable, due to limited segmental hypertrophy or moderate LV dilatation. Thus, quantitative inclusion criteria would be preferable to visual, subjective inclusion criteria. However, the number of discrepancies remained limited.

### 4.4. Sources of Errors Related to the Algorithm

Aside from human errors, how can we try to explain errors related to the algorithm? As a “black box” method with a multilayer nonlinear structure, deep neural networks are often criticized for being nontransparent, and their predictions are not easy to interpret. Yet, it would be better to be able to trust a prediction whose reasons are understandable. High-level feature map visualization of our model would not be very useful for this issue. In contrast, “explainers” such as Grad-CAM [19] or “Class-Selective Relevance Mapping” [21], class-discriminative localization techniques, provide some clues, thanks to the visualization of salient, relevant pixel areas that are the most responsible for image classification prediction. In this study, Grad-CAM heatmaps of class activation showed that the majority of errors were related to the fact that activated pixels in the ground-truth class image were located outside of the left-ventricular region. This was mostly observed in cases of misclassification but also in a few true positive cases. This implies that the correct diagnosis was, thus, the result of chance or that inappropriate features were used to make the correct classification. CNNs follow unintended shortcut strategies, selecting only a few predictive features instead of taking all evidence into account [22]. This can also hint at a problem called hidden stratification [23]; however, error auditing was not able to recognize anomalous patterns in our cases. This observation suggests two types of modifications to be made to our classification process. First, a preliminary segmentation task intended to mask the extracardiac region could be applied, in order to focus the attention of the predictive models on pixels with relevant visual features [7,24]. Furthermore, fine-tuning could be extended to more layers, or another type of CNN network could be tested. However, in this way, complete transparency of the CNN network will not be total; for this reason, Zheng et al. [24] proposed a more simple and straightforward cardiac pathology classification model (logistic regression) with only a few quantitative input features (cardiac volume and LV ejection fraction obtained by deep learning segmentation), returning to the classical, analytical, and explainable method of decision making in medicine.

### 4.5. Limitations

The choice of the target diseases (HCM vs. DCM), which are usually easily distinguishable visually, limits the clinical impact of our study, but it should be reminded that we are only at an early stage in the use of artificial intelligence in this field. Moreover, cardiomyopathies taken into account are only part of this large field of diseases [10]. Non-compaction, overload disease such as amyloidosis or Fabry disease, right-ventricular cardiomyopathy, and other variants were not included here. One can, however, expect good capacities of discrimination for non-compaction with CNN, because of the geometrical characteristics of the hypertrabeculated endocardial contours in this disease. Other overlapping phenotypes such as athlete’s heart and hypertensive cardiomyopathy were not considered in this preliminary work. Only cine-MR was analyzed, which is not enough to allow diagnosis of overload diseases, since T1 mapping and post-gadolinium late enhancement analysis need to be performed as well, as applied in the study of Martini et al. to diagnose amyloidosis [11].

Only one or two selected still images in each cine set were fed to the CNN network, and a 3% significant improvement was obtained by combining diastole and systole (dual input model) instead of looking at diastole or systole solely. This is not how clinical interpretation is done, where multiple cine and other images are used to arrive at the final diagnosis. Taking into account more frames of the cine set should probably improve the classification capability, but this would rely on more sophisticated algorithms (RNN, LSTM) not evaluated in this work. Other methods for time-series feature extraction from the whole cine MR sequence, for example, an apparent flow map [24] or optical flow [25], have been proposed.

Myocardial texture analysis is an interesting further step, able to go beyond visually identifiable structures. Several works using deep learning from native T1-maps or from cine-MR extracted texture features have shown surprising capabilities for discrimination between HCM and hypertrophy related to hypertension [26], between recent and old infarction [27], or to identify DCM [28]. However, it is necessary to first draw a myocardial region of interest to extract texture features, which was not done in our study.

### 4.6. Perspective

This work fits into the perspective of automatic diagnosis in imaging systems, even if the classification task performed here constitutes a fairly simple objective compared to other more difficult issues such as congenital heart disease or the identification of areas of infarction, for example. The learning process to build a model is quite long but the prediction for a given image is almost instantaneous with any type of computer. It becomes, therefore, quite possible to make “online” predictions on imaging devices as the examination proceeds, displaying the diagnosis probability in a corner of the screen. This might be potentially relevant in the early diagnosis of cardiomyopathies. These results could even modulate the examination protocol by proposing subsequent sequences according to the suggested diagnosis. Moreover, a system of acceptance or rejection by the operator would gradually increase the database and, thus, further improve the performance of the algorithm. Beyond these practical aspects, we can expect help and added value for the physician from CNNs in some difficult issues when classifying cardiomyopathies, e.g., when an LVH due to arterial hypertension progresses to dilated CMP, in the differentiation between dilated and non-compaction CMP, or in genetic hypertrophic obstructive or nonobstructive CMP [12].

## 5. Conclusions

In this work, we showed that the implementation of a classical convolutional neural network allows for the classification of normal heart, as well as hypertrophic and dilated cardiomyopathies, from cine-MR images with 98% accuracy. Misclassification mostly occurred when extracardiac regions caught the attention of the network, which may be further improved. Despite several limitations, this study corroborates the excellent computer vision capabilities for automatic diagnosis help in cardiac imaging. Larger, multicenter studies are needed to confirm these encouraging results, which herald the spread of deep learning in cardiac imaging as in other fields of medicine.

## Figures and Tables

**Figure 1 diagnostics-11-01554-f001:**
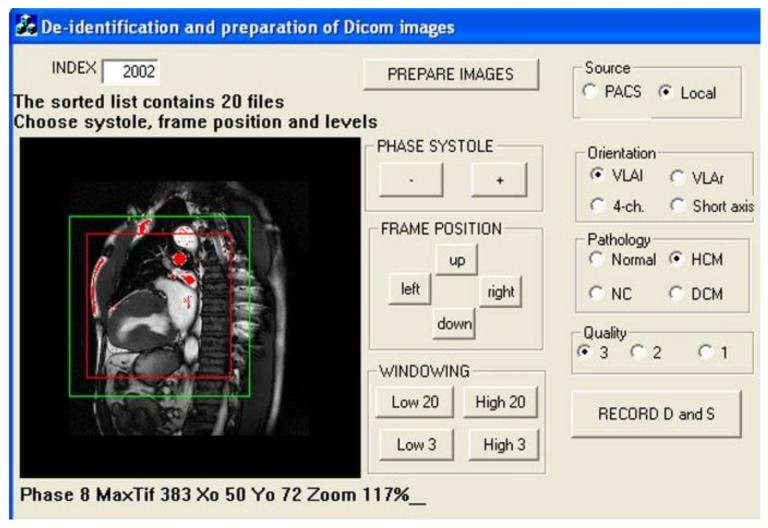
Image preparation: images are cropped to 128 × 128 (red frame) and 160 × 160 (green frame) matrix size by manual displacement of the region of interest on the left ventricle. Gray level is manually adjusted so as to assign the brightest cardiac structures to 255 (1 byte depth per pixel) with help of an ‘over-range blanking’ tool (red area). The four-class orientation label and three-class pathology label are assigned.

**Figure 2 diagnostics-11-01554-f002:**
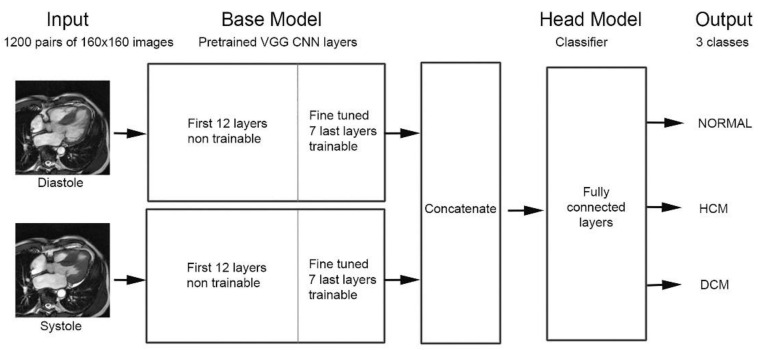
VGG_concat model: diastolic and systolic frames feed two separate pretrained, fine-tuned VGG base models. Feature maps of both outputs are concatenated and supply the fully connected head model providing three (pathology) or four (orientation plane) output classes.

**Figure 3 diagnostics-11-01554-f003:**
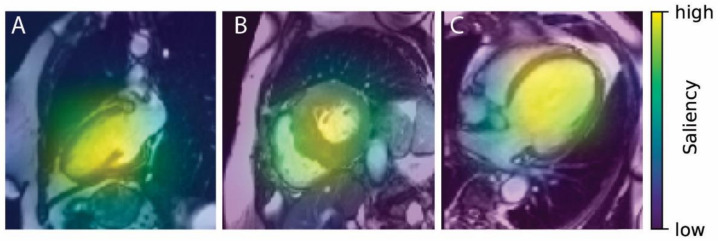
Heatmaps (diastole) corresponding to correct ground-truth class ((**A**) normal, (**B**) HCM, (**C**) DCM).

**Figure 4 diagnostics-11-01554-f004:**
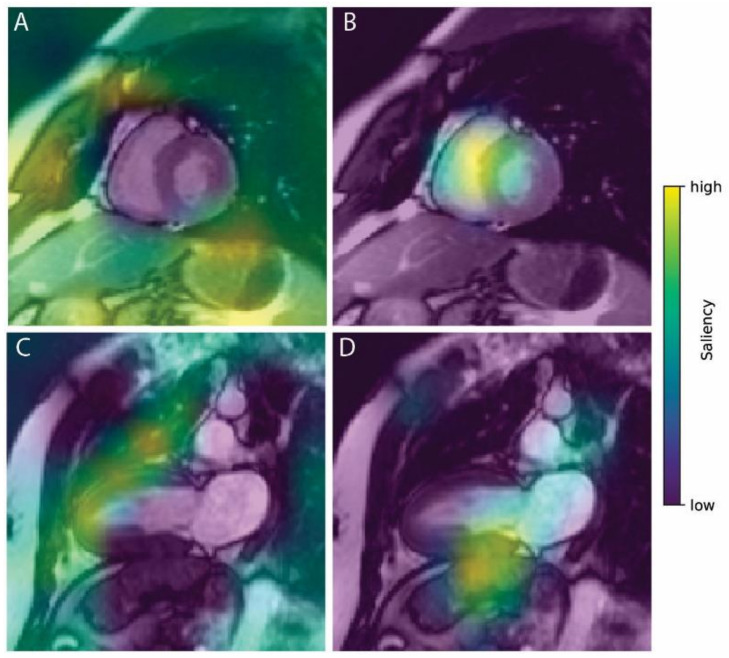
Two examples of misclassified normal heart with erroneous saliency heatmaps. In the ground-truth class images (**A**,**C** on the left), pixels used for classification (yellow-green area) are located outside or at the border of the cardiac region, preventing the algorithm from correctly identifying this class as the correct one. Here, DCM (panel **B**) and HCM (panel **D**) were erroneously selected.

**Table 1 diagnostics-11-01554-t001:** Summary statistics of the cine set included in the study.

	Normal	HCM	DCM	Total	*p*
*n* patients	209	175	150	534	
*n* frames	395	411	394	1200	
Sex (F/M)	148/247	112/299	103/291	363/837	0.0007
Age (years)	45.6 ± 15.8	52.5 ± 17.8	56.4 ± 14.3	51.5 ± 16.7	0.01
VLAl	39	39	66	144	<0.0001
VLAr	106	49	50	205	
4-chamber	132	143	142	417	
Short axis	118	180	136	434	
Systolic time	329 ± 35	346 ± 39	337 ± 29	338 ± 33	0.001

HCM: hypertrophic cardiomyopathy, DCM: dilated cardiomyopathy, VLAl: vertical long axis with left sided apex, VLAr: vertical long axis with right sided apex. *p* denotes the level of statistical significance of differences between pathological groups.

**Table 2 diagnostics-11-01554-t002:** Performance of the two models tested with sixfold cross-validation.

Model	Frames	Classification of Orientation Planes (4 Classes)	Classification of Pathology (3 Classes)
VGG-single	diastole	0.999 ± 0.002 (ns)	0.961 ± 0.011 (*p* = 0.016)
VGG-single	systole	0.998 ± 0.002 (ns)	0.952 ± 0.012 (*p* = 0.0092)
VGG-concat	D + S	0.999 ± 0.002	0.982 ± 0.009

Test accuracy average ± standard deviation of the sixfold cross validation after 100 epochs of training. The value between parentheses denotes the significance level of the difference as compared with the VGG-concat model.

**Table 3 diagnostics-11-01554-t003:** Summed confusion matrices obtained with the sixfold cross-validation training.

VGG-Single Diastole			VGG-Single Systole			VGG-Concat (Diastole + Systole)		
369	4	21	359	26	10	390	3	2
9	396	6	14	394	3	9	400	2
6	1	388	1	0	393	4	0	388
47/1200 misclassified inputs (3.92%)			54/1200 misclassified inputs (4.50%)			22/1200 misclassified inputs (1.83%)		

Summed confusion matrices for classifying pathology, resulting from the sixfold cross-validation for the two models tested. Ground-truth labels (normal, HCM, DCM) are listed vertically, and predicted classes are listed horizontally.

**Table 4 diagnostics-11-01554-t004:** Performance of VGG-single model for systole as a function of the image matrix size.

VGG-Single (S)	Average Validation Accuracy
128 × 128	0.954 ± 0.011 (ns)
160 × 160	0.959 ± 0.009
256 × 256	0.921 ± 0.008 (*p* = 0.0001)
Raw	0.915 ± 0.007 (*p* = 0.0001)

Average ± standard deviation of the 10 last validation accuracies obtained in the validation groups during 100 epochs of training, according to the input matrix size. The value between parentheses denotes the significance level of the difference as compared with the 160 × 160 frame size input.

## Data Availability

The database and code can be made available on reasonable request, after agreement of the Clinical Research Department of our hospital.

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
