# Peer review of "Classification of Cardiomyopathies from MR Cine Images Using Convolutional Neural Network with Transfer Learning"

_diagnostics, 2021, doi:10.3390/diagnostics11091554_

Round 1

Reviewer 1 Report

The authors present the application of neural networks to classify cardiac MR images into three classes (healthy, DCM, HCM). They compare different network architectures, preprocessing steps and hyper-parameters on the network performance. The best performing settings produce a high accuracy for this task on the test data. Studies of this kind, demonstrating the general applicability of deep learning methods in specific medical imaging tasks are justified and valuable. However, there are some major concerns that need to be addressed:

- there are two tasks presented: classifying images into different views (short-axis, long-axis, 4-chamber) and classifying pathology (normal, DCM, HCM). It is not clear to me, whether both tasks were performed by a single network or whether separate networks were trained. Also I don't see why the first task is relevant in this context at all. 
- it is surprising that images of different views are classified by the same network, this makes learning relevant features from images potentially much harder as it increases variability unrelated to any disease. This is not true if different views are already indicative of the disease (e.g. by having some views excluded for certain disease (sub-)types, see l.88-90). In this case, however, bias is introduced and the performance of the model on data that is not manually pre-selected in the same way is probably overestimated.
- as presented in 2.1 the study population is not homogeneous in each class (e.g. regarding gender and age). This may also lead to overestimation of the model performance in a realistic setting. You can consider trying stratification to see how this affects the model.
- to create ground truth labels not only the images fed to the network but all available information should be used (including other MR series, measurements, medical records, ...)
- it is great that you compare model performance to human performance. Keep in mind, however, that for clinical application it is irrelevant how well humans perform when imitating the task of the network (i.e. looking at single pre-processed images). Instead classification accuracy needs to be compared to the accuracy of humans following standard clinical procedure for classification.
- when you use cross-validation on the full dataset to select settings and tune hyper-parameters and later use a hold-out set from the same full dataset, the results are not reliable. Even though training and validation does not happen on the hold-out set, information from these images already leaked by influencing the selection of hyper-parameters.
- phrasing the saliancy maps as "where the network looks" is a bit misleading. The network is "looking" at the full image in any case. The highlighted regions are the ones that contain features that influenced the classification most strongly. So it is more like what "caught the attention of the network". 
- making a correct predictions while the most saliant regions are outside of the heart does not indicate that it is correct by chance (l.224f, l.299f). It can also mean that unintended features were used to make the correct classification. These are sometimes called short-cut features (https://arxiv.org/abs/2004.07780) it can also hint to a problem called "hidden stratification" (https://arxiv.org/abs/1909.12475).
- for reproducibility and reuse of the results as much of the data, code and models should be published alongside the article. When data can not be shared due to ethical and regulatory reasons, this is acceptable but not "owing to planned future publication" (l.382). Also intermediate results like the accuracy of all tested networks can still be published as supplementary data.

minor:
- the background section in the abstract does not provide background but starts directly with the aim of the work
- the introduction lacks references, e.g. for the first section
- it it not clear whether percentage improvements should be interpreted as relative or absolute improvements. I.e. an improvement of a 80% accuracy by 10% can be 88% (relative) or 90% (absolute).
- in section 3.4 how big is the overlap of the mis-classifications by the network and the humans (and also between the humans). Did they mis-classify the same or different images?
- Fig. 3 caption: "true positive" is not meaningful if the classification task is not binary
- the claimed generalizability regarding vendor, etc. has not been demonstrated systematically

Author Response

The authors present the application of neural networks to classify cardiac MR images into three classes (healthy, DCM, HCM). They compare different network architectures, preprocessing steps and hyper-parameters on the network performance. The best performing settings produce a high accuracy for this task on the test data. Studies of this kind, demonstrating the general applicability of deep learning methods in specific medical imaging tasks are justified and valuable. However, there are some major concerns that need to be addressed:

- there are two tasks presented: classifying images into different views (short-axis, long-axis, 4-chamber) and classifying pathology (normal, DCM, HCM). It is not clear to me, whether both tasks were performed by a single network or whether separate networks were trained.

In section 2.4 it has been clarified that : “Specific models for orientation plane and for pathology were trained through …

Also I don't see why the first task is relevant in this context at all. 

The classification of the imaging planes was not used for further steps in this work but it was interesting to evaluate capability of imaging plane identification (which appear to be excellent), as a secondary goal of the study. Other authors, using ultrasound imaging, proceeded in the same way  (ref 7 & 8). Moreover, this classification is an important preliminary step for other kind of tasks such as heart chamber segmentation.

- it is surprising that images of different views are classified by the same network, this makes learning relevant features from images potentially much harder as it increases variability unrelated to any disease. This is not true if different views are already indicative of the disease (e.g. by having some views excluded for certain disease (sub-)types, see l.88-90). In this case, however, bias is introduced and the performance of the model on data that is not manually pre-selected in the same way is probably overestimated.

Thank you for this important comment. Yes, we agree that it is surprising that images of different orientation led to efficient classification of diseases and this enhances the strength of the model even if –as stated in the method section – images were selected as to include features visually suggestive of the disease. Such was not necessary for dilated cardiomyopathies because typical pattern is visible in almost all imaging views but this was necessary for some segmental hypertrophic cardiomyopathies (HCM) because myocardial wall thickening may be restricted to few selective views or orientation planes. Diagnosis of HCM in such cases can only be ascertained if the relevant view is seen by either human eye or by the algorithm; diagnosis will be wrong if a suggestive feature is not present in the analyzed image. Nevertheless, some views were NOT excluded for certain disease.

This consideration raises the tricky question of performing analysis either on a ‘imaging slice’ basis (which has been done here and which is usually performed in this kind of work) or on a ‘patient’ basis, relying on the whole set of images of the patient’s exam. BUT, on one hand implementation of the second option is quite more difficult and reduces the number of observations (one patient instead of several images) which is deleterious for CNN training ; on the other hand relevant diagnosis may be correctly suggested even if only one single image in the imaging set is typical for a given condition (for example segmental septal hypertrophy involving only one imaging slice) and thus the diagnosis may rely on classification of a single imaging plane (such as done in our study).

In the “limitations” section of the discussion we have specified :  “This is not how clinical interpretation is done, where multiple cine and other images are used to arrive at the final diagnosis …Taking into account more frames of the cine set should probably improve the classification capability but relies on more sophisticated algorithms…”

- as presented in 2.1 the study population is not homogeneous in each class (e.g. regarding gender and age). This may also lead to overestimation of the model performance in a realistic setting. You can consider trying stratification to see how this affects the model.

Following your advice, the model was trained selectively for male or female and selectively for restricted ranges of age and the following sentences were added in the section results 3.2 :

Since the study population is not homogeneous in each class (e.g. regarding gender and age), stratified analysis was performed. According to sex, accuracy for the held-out test group was 0.932 for male (57% of cases) and 0.883 for female (43%). According to age, we found 0.889 in patients <46-year-old (32% of cases), 0.879 in patients 45–62-year-old (33%) and 0.907 in patients > 62-year-old (35%). These differences are interpreted as related to the number of cases studied in each sub-group. (l198)

- to create ground truth labels not only the images fed to the network but all available information should be used (including other MR series, measurements, medical records, ...)

According to the design of this study, after deidentification, several items extracted from the Dicom tags were automatically written in the label file, including, year of examination, sex, age, magnetic field strength, name of the scanner, pixel spacing, cardiac phase, trigger time, highest pixel value and image position. Only 3 items were entered manually by the radiologist during the image preparation step:  image plane orientation, type of disease (especially healthy, hypertrophic or dilated cardiomyopathy) and image quality. Additional data such as clinical parameters or measurements available in the exam report would have been very interesting for extended analysis but this means a tedious supplementary work and we thought that the inclusion of all those parameters was not mandatory for our initial purpose.

- it is great that you compare model performance to human performance. Keep in mind, however, that for clinical application it is irrelevant how well humans perform when imitating the task of the network (i.e. looking at single pre-processed images). Instead classification accuracy needs to be compared to the accuracy of humans following standard clinical procedure for classification.

Yes, we fully agree with your comment, which highlights the difference between an experimental study and the real-world and send us back to our previous remark related to ‘single image’ classification (image based) vs ‘full images set’ classification (patient based) which relies on a much more demanding study design. If human classification would have been performed following standard clinical procedure (according to the whole image set), almost no error would have been done by humans.

- when you use cross-validation on the full dataset to select settings and tune hyper-parameters and later use a hold-out set from the same full dataset, the results are not reliable. Even though training and validation does not happen on the hold-out set, information from these images already leaked by influencing the selection of hyper-parameters.

Thank you for pointing out this question, which led us to clarify in the text (material and Methods section 2.4): “Model trimming was limited to few alterations of the number of epochs and of dropout values in the head model. Thus, since adjustments of hyperparameters were minimal, information leaks were almost absent and cross validation could be considered valid (nested cross validation not mandatory)”.

- phrasing the saliency maps as "where the network looks" is a bit misleading. The network is "looking" at the full image in any case. The highlighted regions are the ones that contain features that influenced the classification most strongly. So it is more like what "caught the attention of the network".

We took this remark into account by changing the turn of the phrase : “In the situation where misclassification occurred, heatmaps inspection revealed that the main source of error (half of the cases: 11/22) was related to the fact that extracardiac location caught the attention of the network, as illustrated in figure 4.” (l252)

“networks looks outside of the cardiac region” was also changed to “Misclassification mostly occurred when extracardiac regions caught the attention of the network » in the conclusion section.

- making a correct prediction while the most saliant regions are outside of the heart does not indicate that it is correct by chance (l.224f, l.299f). It can also mean that unintended features were used to make the correct classification. These are sometimes called short-cut features (https://arxiv.org/abs/2004.07780) it can also hint to a problem called "hidden stratification" (https://arxiv.org/abs/1909.12475).

We have reported this interesting information provided by the articles that you have mentioned in both corresponding sections of the text and we have added the two references:

  • most misclassifications occurred when extracardiac location caught the attention of the network (abstract)
  • “This implies that the correct diagnosis was thus the result of chance” was removed and replaced by “…was correctly performed due to unintended features” (l252)
  • “or that inappropriate features were used to make the correct classification. CNNs follow unintended “shortcut” strategies, selecting only a few predictive features instead of taking all evidence into account [Geirhos]. It can also hint to a problem called hidden stratification [Oakden] (but error auditing was not able to recognize anomalous patterns in our cases)” (l332)

- for reproducibility and reuse of the results as much of the data, code and models should be published alongside the article. When data can not be shared due to ethical and regulatory reasons, this is acceptable but not "owing to planned future publication" (l.382). Also intermediate results like the accuracy of all tested networks can still be published as supplementary data.

"owing to planned future publication" has been removed.

Minor:

- the background section in the abstract does not provide background but starts directly with the aim of the work

At the beginning of the background section, we have added :  “Automatic classification of various type of cardiomyopathies is desirable but has never been performed by CNN method”.

- the introduction lacks references, e.g. for the first section

3 references have been added in the first section of the introduction

  • Litjens G, Ciompi F, Wolterink JM et al. State-of-the-Art Deep Learning in Cardiovascular Image Analysis. JACC Cardiovasc Imaging. 2019 Aug;12(8 Pt 1):1549-1565. doi: 10.1016/j.jcmg.2019.06.009.
  • Liu X, Faes L, Kale AU et al. A comparison of deep learning performance against health-care professionals in detecting diseases from medical imaging: a systematic review and meta-analysis. Lancet Digit Health. 2019 Oct;1(6):e271-e297. doi: 10.1016/S2589-7500(19)30123-2.
  • Martin-Isla C, Campello VM, Izquierdo C et al. Image-Based Cardiac Diagnosis With Machine Learning: A Review. Front Cardiovasc Med. 2020 Jan 24;7:1. doi: 10.3389/fcvm.2020.00001.

- it is not clear whether percentage improvements should be interpreted as relative or absolute improvements. I.e. an improvement of a 80% accuracy by 10% can be 88% (relative) or 90% (absolute).

Percentage improvement is expressed as absolute value (not relative). This precision has been mentioned in the text.

- in section 3.4 how big is the overlap of the mis-classifications by the network and the humans (and also between the humans). Did they mis-classify the same or different images?

This clarification has been added in the text : “The misclassification made by the two human observers concern the same image in 1/15 case only. Similarly, the errors made by the algorithm concern the same image as for human observers in only one case (for both readers). In the event of a discrepancy between the algorithm and the human, the latter is right in a little more than half of the cases.” (l225)

- Fig. 3 caption: "true positive" is not meaningful if the classification task is not binary

We modified the sentences

  • as “Heat maps corresponding to correct diagnosis...” (fig 3 legend, l248)
  • and in section 3.5: “When the correct class is identified, …”

- the claimed generalizability regarding vendor, etc. has not been demonstrated systematically

We turned the sentence into: “the model seems to be well generalizable, for example…” (l293)

Reviewer 2 Report

This manuscript suggests the use of transfer learning-based convolutional neural network for the classification of cardiomyopathies from cardiac magnetic resonance (CMR) images.

Although the technical and “biological” limitations of this work, precisely pointed out by the authors, the topic is timely and relevant in the clinical landscape.

The manuscript could be improved in line with the following comments:

  • Please speculate about the potential relevance of this findings in the early diagnosis of cardiomyopathies.
  • The literature in this field is growing day by day. Recently, new updates have been published and deserve a discussion.
  • Please define the abbreviations in the abstract and every time they are used for the first time in the text.
  • English language and style are fine, but minor spell check is required.

Author Response

This manuscript suggests the use of transfer learning-based convolutional neural network for the classification of cardiomyopathies from cardiac magnetic resonance (CMR) images.

Although the technical and “biological” limitations of this work, precisely pointed out by the authors, the topic is timely and relevant in the clinical landscape.

The manuscript could be improved in line with the following comments:

- Please speculate about the potential relevance of this findings in the early diagnosis of cardiomyopathies.

Thanks for your comment. As you suggested we added the following sentence in the perspective section “This might be potentially relevant in the early diagnosis of cardiomyopathies.” L383

- The literature in this field is growing day by day. Recently, new updates have been published and deserve a discussion.

We fully agree with your comment. 3 recent references including the exhaustive review of Martin-Isla were added in the introduction.

  • Litjens G, Ciompi F, Wolterink JM et al. State-of-the-Art Deep Learning in Cardiovascular Image Analysis. JACC Cardiovasc Imaging. 2019 Aug;12(8 Pt 1):1549-1565. doi: 10.1016/j.jcmg.2019.06.009.
  • Liu X, Faes L, Kale AU et al. A comparison of deep learning performance against health-care professionals in detecting diseases from medical imaging: a systematic review and meta-analysis. Lancet Digit Health. 2019 Oct;1(6):e271-e297. doi: 10.1016/S2589-7500(19)30123-2.
  • Martin-Isla C, Campello VM, Izquierdo C et al. Image-Based Cardiac Diagnosis With Machine Learning: A Review. Front Cardiovasc Med. 2020 Jan 24;7:1. doi: 10.3389/fcvm.2020.00001.

- Please define the abbreviations in the abstract and every time they are used for the first time in the text.

This has been done.

- English language and style are fine, but minor spell check is required.

The text has been revised once more by an English speaker from our institution

Round 2

Reviewer 1 Report

The authors responded to all of my concerns and improved the manuscript in many regards.
Some minor concerns remain:
- your point about prediction on image vs patient level is very interesting and valid. Still, the clinically relevant level of prediction is the patient level. So one possibility to further evaluate and validate the results of your final trained models would be:
   - make predictions on all images of some (previously unseen) patients
   - see how consistent the classification is among images
   - ideally all images would be classified the same
   - however this can not be expected in case of HCM, where only some images show the disease
   - in this case it is interesting if the images correctly classified as HCM are indeed the ones showing the disease
   - similarly for healthy patients: are some images classified as HCM (maybe just because it is a slice that often shows myocardial wall thickening, if it exists)
- this kind of additional evaluation would massively help in interpreting the model performance and generalizability and would not require much additional data
- if only "one single image in the imaging set is typical for a given condition" manually pre-selecting this image is no improvement compared to manually doing the classification
- "nested cross-validation not mandatory", I understand your argument but how can you be sure that "leaks were almost absent", did you check for that? How?
- will data/models/code be shared? In my opinion sharing material to the extent possible is mandatory for publication.

Author Response

The authors responded to all of my concerns and improved the manuscript in many regards. Some minor concerns remain:

- your point about prediction on image vs patient level is very interesting and valid. Still, the clinically relevant level of prediction is the patient level. So one possibility to further evaluate and validate the results of your final trained models would be:
  - make predictions on all images of some (previously unseen) patients

According to your suggestion, the model obtained with the 1200 cases of our previous work on cardiomyopathies was stored as a .h5 file.

New normal, HCM and DCM cases, collected since April 2021 (in the view of a future study) were used as previously unseen inputs (795 pairs of diastolic and systolic frames). Predictions, based on the .h5 file, performed on this new series of data, provide 33 misclassifications/795 cases, i.e. accuracy is 95,8% (see the listing below)

  - see how consistent the classification is among images

  - ideally all images would be classified the same

The overall error rate is 33/795 (i.e. 4.15%) on a frame basis. Errors concerned two frames from the same patient on 3 occasions and a single frame from a given patient in other cases.

If all frames in each patient would contain the typical features of the same pathological class (e.g. hypertrophy), then the overall error rate per patient would be 4.15% x the number of frame per patient , i.e. 4.15 x 2.25 = 9.3% on a patient basis.

Actually, all frames of a given patient do not always correspond to the same pathological class. The typical example is “localized, segmental septal hypertrophy” leading to 1 or 2 images with HCM class, whereas other images may be classified as normal. Thus, the error rate on a patient’s basis should be less than 9.3%, perhaps in the range 5-8%. The problem is that the initial design of our study was not made to assess the misclassification on a patient’s basis but on a frame basis.

  - however this can not be expected in case of HCM, where only some images show the disease

  - in this case it is interesting if the images correctly classified as HCM are indeed the ones showing the disease

Images correctly classified as HCM always showed the disease. HCM were misclassified as ‘normal’ in 7 cases and as ‘DCM’ in 3 cases.

  - similarly for healthy patients: are some images classified as HCM (maybe just because it is a slice that often shows myocardial wall thickening, if it exists)

Healthy patients were misclassified as HCM in 12 cases. As shown in the list below, moderate segmental hypertrophy was only seen in 2 out of 12 of these ‘normal’ case.

- this kind of additional evaluation would massively help in interpreting the model performance and generalizability and would not require much additional data
- if only "one single image in the imaging set is typical for a given condition" manually pre-selecting this image is no improvement compared to manually doing the classification

Main results of the supplementary analysis, previously unseen by the model, have been added in the text (Cardiomyopathies-revised3.docx, attached) :

  • Finally, a complementary series of 795 input, previously unseen by the model was tested separately. (line 165 in the method section)
  • Finally, the additional analysis carried out on 795 supplementary inputs, never seen before by the model, showed 33/795 errors, i.e. an accuracy of 958 (line 205 in the result section)

Listing of the 33 misclassified cases from the supplementary analysis performed on a series of 795 new cases previously unseen by the model

Frame          misclassification       Nota / explanation

1236                 0 → 2                   error that cannot be explained by visual analysis

1217                 0 → 2                   error that cannot be explained by visual analysis

1246                 0 → 1                   error that cannot be explained by visual analysis

1359                 0 → 1                   sub-aortic septal thickening

1470                 0 → 1                   error that cannot be explained by visual analysis

1498                 2 → 1                   error that cannot be explained by visual analysis

1506                 0 → 1                   error that cannot be explained by visual analysis

1543                 0 → 1                   error that cannot be explained by visual analysis

1544                 0 → 1                   idem (same patient as 1543)

1563                 2 → 0                   error that cannot be explained by visual analysis

1567                 0 → 1                   error that cannot be explained by visual analysis

1573                 0 → 1                   error that cannot be explained by visual analysis

1616                 2 → 0                   error that cannot be explained by visual analysis

1620                 2 → 0                   error that cannot be explained by visual analysis

1657                 0 → 1                   moderate thickening of the anterior wall

1739                 1 → 0                   septal thickening is moderate

1790                 2 → 0                   error that cannot be explained by visual analysis

1839                 0 → 1                   error that cannot be explained by visual analysis

1848                 2 → 1                   error that cannot be explained by visual analysis

1861                 1 → 0                   frank labeling error (is DCM and not HCM)

1905                 1 → 2                   error that cannot be explained by visual analysis

1970                 1 → 2                   inferior wall thickening is moderate

2178                 1 → 0                   limited sub-aortic septal thickening

2193                 2 → 1                   error that cannot be explained by visual analysis

2245                 1 → 0                   error that cannot be explained by visual analysis

2294                 2 → 1                   error that cannot be explained by visual analysis

2295                 2 → 1                   idem (same patient as 2294)

2344                1 → 2                    error that cannot be explained by visual analysis

2400                1 → 0                    error that cannot be explained by visual analysis

2798                1 → 0                   error that cannot be explained by visual analysis

2872                1 → 0                   error that cannot be explained by visual analysis

2927                0 → 1                   error that cannot be explained by visual analysis

2928                0 → 1                   idem (same patient as 2927)

9/33 occurrences with limited features (questionable)

Two misclassified frames from the same patient on 3 occasions.

- “nested cross-validation not mandatory”, I understand your argument but how can you be sure that “leaks were almost absent”, did you check for that? How?

No, this was not formally checked. We modified our sentence as “Model trimming was limited to few alterations of the number of epochs and of dropout values in the head model. Thus, since adjustments of hyperparameters were minimal, information leaks should in principle be almost absent, so that nested cross-validation was not perfomed”.

- will data/models/code be shared? In my opinion sharing material to the extent possible is mandatory for publication.

Concerning the issue of sharing data, models and software, the Clinical Research Department of our hospital is not in favor of it a priori; the general principle being that the data must not leave the hospital.

This principle could probably be reviewed within the framework of the present work and this is why we indicated the mention: “database and code can be made available on reasonable request, after agreement of the Clinical Research Department of our hospital” in the Data Availability Statement.